# Technical Note: Streamflow Seasonality using Directional Statistics

Wouter R. Berghuijs[1], Kate Hale[2], Harsh Beria[3,4]

[1]Department of Earth Sciences, Free University Amsterdam, Amsterdam, the Netherlands
[2]Department of Geography, University of British Columbia, Vancouver, Canada
[3]WSL Institute for Snow and Avalanche Research SLF, Davos, Switzerland
[4]Department of Civil, Environmental and Geomatic Engineering ETH Zurich, Zurich, Switzerland

*Correspondence to*: Wouter R. Berghuijs (w.r.berghuijs@vu.nl)

**Abstract.** Hydrological fluxes typically vary across seasons, with several existing metrics available to characterize their seasonality. These metrics are beneficial when many catchments across diverse climates and landscapes are studied concurrently. Here, we present directional statistics to characterize streamflow seasonality, capturing the timing of streamflow (center of mass timing) and the strength of its seasonal cycle (center of mass concentration). We show that directional statistics are mathematically more robust than several widely used metrics to quantify streamflow seasonality. We extend the application of directional statistics to analyse seasonality in other hydrological fluxes, including precipitation, evapotranspiration, and snowmelt, and we introduce a trend analysis framework for both the timing and strength of seasonal cycles. Using an Alpine catchment (Dischma, Switzerland) as a testbed for this methodology, we identify a shift in the streamflow center of mass to earlier in the year and a weakening of the seasonal cycle. Additionally, we apply directional statistics to streamflow data from 11,118 European catchments, highlighting its utility for large-scale hydrological analyses. The introduced metrics, leveraging directional statistics, can improve our understanding of streamflow seasonality and associated changes, and can also be used to study the seasonality of other environmental fluxes, within and beyond hydrology.

## 1 Introduction

Most rivers have distinct seasonal variations in streamflow, which tend to affect floods, droughts, water resources, and ecosystems (Poff et al., 1997; Sivapalan et al., 2005; Berghuijs et al., 2014; Knoben et al. 2018; Blöschl et al., 2017; Patil and Shulmeister, 2023). Anthropogenic climate change has influenced streamflow seasonality globally with changes that are often especially pronounced in snow-affected regions (Wang et al., 2024). In such regions, climate warming acts to shift precipitation phase from snowfall towards rainfall, reduces snowpacks, and causes earlier snowmelt, often leading to earlier streamflow (Barnett et al., 2005; Luce et al., 2009; Wang et al., 2024; Berghuijs and Hale, 2025). However, shifts in streamflow seasonality can also be substantial in largely snow-free environments (Wasko et al., 2021; Chalise et al., 2021; Wang et al., 2024).

Quantitative summaries of streamflow seasonality across space and/or time typically rely on analyses related to select streamflow characteristics. For example, streamflow records can be described by their multi-year mean monthly flows,

sometimes normalized by the mean annual flow rate (Pardé, 1933). Such quantitative descriptions can help understand streamflow regime patterns and their changes but mean monthly flows continue to be time series. Therefore, if one aimed to, further steps are required to characterize streamflow seasonality across many catchments simultaneously, for example, create groups of similar flow regimes (e.g., Haines et al., 1988; Berghuijs et al., 2014; Knoben et al., 2018).

Singular metrics capturing streamflow timing characteristics can be calculated to target when a particular amount of flow has passed since the start of the (water) year allowing for comparisons of seasonal flow regimes across many catchments and evaluation of flow regime shifts. For example, the *half-flow date* marks the time elapsed from the beginning of the water year to when half of the annual streamflow is reached (Court et al., 1962):

$$t_{\frac{1}{2}} = \frac{\int_{t=0}^{t=t_{\frac{1}{2}}} Q(t)\, dt}{\int_{t=0}^{t=1} Q(t)\, dt} = 0.5 \qquad \text{(Eq. 1)}$$

where $t_{\frac{1}{2}}$ is the half flow date [T], $t$ is the time (fraction) since the start of the water year [T], and $Q(t)$ is the mean flow rate at time $t$ [L$^3$/T or L/T] (averaged over the studied period). $t_{\frac{1}{2}}$ represents the date corresponding to the median of the cumulative flow distribution. Alternatively, a *center of mass* represents the point in time when the weighted position vectors of the streamflow relative to this moment (i.e., streamflow multiplied by the time difference to the center of mass), sum to zero:

$$t_{\hat{Q}} = \frac{\int_{t=0}^{t=1} (t \cdot Q(t))\, dt}{\int_{t=0}^{t=1} Q(t)\, dt} \qquad \text{(Eq. 2)}$$

where $t_{\hat{Q}}$ is the center of mass [T], $t$ is the time (fraction) since the start of the water year [T], and $Q(t)$ is the flow rate at time $t$ [L$^3$/T or L/T] (averaged over the studied period). $t_{\hat{Q}}$ represents the date corresponding to the mean of the cumulative flow distribution. The concepts of half-flow date and center of mass are widely used (e.g., Court 1962; Stewart et al., 2005; Yang et al., 2007; Regonda et al., 2005; Hodgkins et al., 2003; Luce et al., 2009; Clow et al., 2010; Kormos et al., 2016; Renner & Bernhofer, 2011; Han et al., 2024; Gnann et al., 2021; Chen et al., 2023; Botterill and McMillan, 2023; Almagro et al., 2024). However, it is important to note that while these terms are often used interchangeably, Eq. 1 and Eq. 2 yield different values when the seasonal flow regime is even slightly skewed, which invariably occurs. Additionally, it is important to recognize that seasonal streamflow patterns within water years occur in unbounded timeseries (i.e. annual cycles), meaning two dates can be considered adjacent even if they fall on opposite ends of the time series. For example, if the water year begins on October 1st, September 30th marks the end of the time series but is also directly adjacent to October 1st.

The strength of streamflow seasonality can also be calculated using a variety of metrics (also commonly applied to precipitation). These metrics quantify the degree of (monthly) variability without considering the sequence of these mean monthly values. For example, a commonly used seasonality index is (Walsh and Lawler, 1981; Eisner et al., 2017):

$$I_{S1} = \frac{1}{Q_a} \sum_{n=1}^{12} \left| Q_n - \frac{Q_a}{12} \right| \tag{Eq. 3}$$

where $I_{s1}$ is the seasonality index [-], $Q_a$ is the mean annual streamflow sum [$L^3$/T or L/T], and $Q_n$ is the mean monthly streamflow sum [$L^3$/T or L/T] for month *n*. Alternatively, the strength of seasonality can be calculated using (Oliver 1980; Han et al., 2024):


$$I_{S2} = \frac{\sum_{n=1}^{12}(Q_n^2)}{Q_a^2} \tag{Eq. 4}$$

or using Apportionment Entropy (Feng et al., 2013; Wang et al., 2024):

$$I_{S3} = - \sum_{n=1}^{12} \left( \frac{Q_n}{Q_a} \log_2 \left( \frac{Q_n}{Q_a} \right) \right) \tag{Eq. 5}$$

Eqs. 3-5 are straightforward to compute but lose valuable information from the (mean monthly) time series, as they ignore the sequence of mean monthly flow rates.


A method that considers both the periodic nature of a mean seasonal streamflow cycle and the sequence of its flow rates is the description of streamflow seasonality using a sine function (e.g., Marvel et al., 2021; Gnann et al., 2020):

$$Q(t) = \overline{Q} \left( 1 + \delta_Q \sin \left( \frac{2\pi(t - \phi_Q)}{Z} \right) \right) \tag{Eq. 6}$$

$Q(t)$ is the flow rate [$L^3$/T or L/T] at time *t*, $\overline{Q}$ is the mean streamflow rate [$L^3$/T or L/T], $\delta_Q$ is a (dimensionless) streamflow
seasonality, $\phi_Q$ is the phase [T] of seasonal streamflow, and *Z* is the period of interest [T] set at 1 year. A sine function is most effective for variables with an annual cycle that closely aligns with a sine curve, such as temperature seasonality (e.g., Stine et al., 2009; Berghuijs and Woods, 2016; Marvel et al., 2021). However, in our experience, most streamflow seasonality distributions deviate substantially from a sine curve (e.g., see Fig. 5 of Knoben et al., 2018).

Here we discuss the use of directional statistics for expressing streamflow seasonality. Directional statistics (Mardia and Jupp, 2000) allow for consideration of the cyclical nature and the sequence of streamflow and can be used for non-sinusoidal time series (e.g. snowmelt). Directional statistics have been widely used to characterize the seasonality of extreme flows and extreme precipitation (e.g., Burn et al., 1997; Young et al., 2000; Merz & Blöschl, 2003; Laaha and Blöschl, 2006; Dhakal et al., 2015; Villarini, 2016; Blöschl et al., 2017; Berghuijs et al., 2016, 2019; Floriancic et al., 2021; Chagas et al., 2022). Recent
developments show these can be adapted to represent the overall intra-annual distribution of streamflow (Jiang et al., 2022; Nan and Tian, 2024; Hanus et al., 2024). Such an approach parallels applications in other scientific fields (e.g., image processing), where centers of mass are calculated for unbounded environments (Bai and Breen, 2008). We show how directional statistics enables simultaneous characterization of both the timing (*center of mass timing*) and strength (*center of mass concentration*) of the seasonal streamflow cycle. We compare these directional statistics with widely used metrics (e.g.,

Eqs. 1-5) and discuss how directional statistics are often (mathematically) more robust. To illustrate potential applications, we use this approach to quantify the seasonality of different hydrological fluxes in an Alpine catchment. Additionally, we apply directional statistics to streamflow timeseries across 11,117 catchments in Europe and quantify trends in the timing and concentration of the seasonal cycle for these catchments.

## 2. Seasonality using directional statistics definitions

The concept of center of mass is not confined to hydrology. It is widely used to determine the average (spatial) position of all mass in a system or object and has many applications in, for example, engineering, astronomy, and biomechanics and can be adapted to a temporal framework. We discuss the seasonal flux equivalent of the center of mass, representing the distribution of a flux in time rather than in space, and account for the cyclical nature of seasonal cycles.

Directional statistics can express the *center of mass timing* of streamflow, $t_{\hat{Q}}$ [T]:

$$t_{\hat{Q}} = \frac{\text{atan2}(\overline{y}, \overline{x})}{2\pi} \qquad (\text{Eq.}\,7)$$

and its *concentration, R* [dimensionless]:

$$R = \sqrt{\overline{x}^2 + \overline{y}^2} \qquad (\text{Eq.}\,8)$$

where the cosine and sine components of streamflow are:


$$\overline{x} = \frac{1}{\int_{t=0}^{t=1} Q(t)\mathrm{d}t} \int_{t=0}^{t=1} \big(\cos(2\pi t)\, Q(t)\big)\mathrm{d}t \qquad (\text{Eq.}\,9)$$

$$\overline{y} = \frac{1}{\int_{t=0}^{t=1} Q(t)\mathrm{d}t} \int_{t=0}^{t=1} \big(\sin(2\pi t)\, Q(t)\big)\mathrm{d}t \qquad (\text{Eq.}\,10)$$

Here $t_{\hat{Q}}$ is the *center of mass timing* expressed as a fraction of a year [T] compared to the start of a (water) year, $t$ is the time (fraction of year) since the start of the (water) year [T], and $Q(t)$ is the flow rate at time $t$ [L$^3$/T or L/T]. Numerical implementation of the above equations requires considering the time interval at which data is provided (e.g., hourly, daily,

weekly, See Appendix A1 for discrete form). However, unlike the aforementioned methods (Eqs 1-5), no further time averaging of data is required.

The center of mass timing ($t_{\hat{Q}}$) represents the unique timing of streamflow concentration within a year, ensuring that the streamflow distribution, weighted by its relative position (streamflow multiplied by the time offset from the center of mass),

sums to zero. This is equivalent to the center of mass provided in Eq. 2, except that $t_{\hat{Q}}$ considers the periodic nature of a seasonal streamflow pattern.

$R$ indicates the strength of mass concentration at that time. An $R$-value close to 1 indicates all streamflow occurs during one isolated moment in the year, whereas $R=0$ indicates that this mass is symmetrically distributed throughout the year. In the latter case, the center of mass timing would remain undefined (indicating no seasonality), but this would be extremely rare using real-world streamflow data. The (directional) variance of $t_{\hat{Q}}$ is: $\mathrm{Var}(t_{\hat{Q}}) = 1 - R$ and the (directional) standard deviation equals $\sigma_{t_{\hat{Q}}} = \sqrt{2\ln(R^{-1})}$. In this technical note, we report the concentration (Eq. 8), not the variance or standard deviation.

To illustrate use and interpretation of directional statistics, we calculate the center of mass timing and associated concentrations for three example catchments (Fig. 1). The seasonal hydrographs of these catchments show distinct patterns (Fig. 1a-c). The Tiebel River at Himmelberg (Austria) drains a 40 km² pre-alpine catchment with most of the flow originating from over 40 springs. These springs are fed by the groundwater flow out of the neighbouring Gurktal and remain almost constant throughout the year (largely unaffected by snowmelt and major rainfall). Consequently, its streamflow exhibits minimal seasonality (Fig. 1a). The Alfenz River at Klösterle (Austria) drains a 66 km² alpine catchment with a seasonal hydrograph with spring melt (Fig. 1b). The Horndrup River at Sortholmvej (Denmark) drains a 5.5 km² agricultural and energy-limited catchment with relatively consistent rainfall throughout the year (Bastrup-Birk and Gundersen, 2004) and thereby experiences winter-dominated streamflow (Fig. 1c). The calculated centers of mass timings (here expressed as the fraction of the year since January 1) and concentration make these intra-annual variations in streamflow quantitative (Fig. 1d-f).

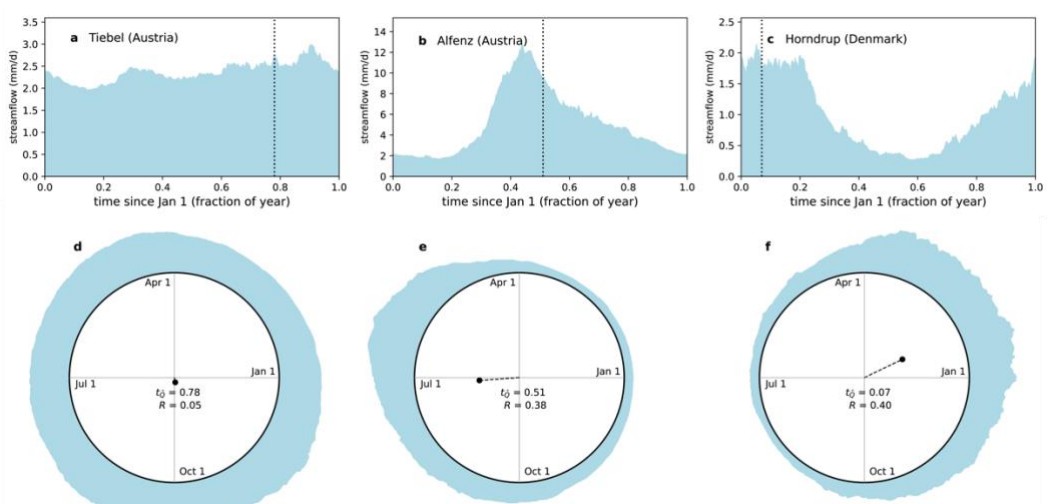

**Figure 1: Seasonal hydrographs and streamflow seasonality for three example catchments. The top row shows the multi-year mean daily streamflow and center of mass for the Tiebel River at Himmelberg (Austria) (a), Alfenz River at Klösterle (Austria) (b), and Horndrup River at Sortholmvej (Denmark) (c). The bottom row presents the streamflow mass distribution using directional statistics, highlighting the center of mass timing (fraction of a year) and concentration (dimensionless). Streamflow data are obtained from the EStreams database (do Nascimento et al., 2024).**

`

## 3. Robustness of directional statistics

Suggesting directional statistics for analysing streamflow seasonality may seem redundant, considering the many already existing metrics (e.g., Eqs. 1-5). However, directional statistics consider the periodic nature and sequence of mean seasonal flow rates, regardless of the analysis start date, and can be used for non-sinusoidal time series. It thereby can have several advantages over commonly used methods used to summarize seasonality. We will illustrate this using simple examples, demonstrated below. We acknowledge that no single metric can fully replace all others, as the choice of metrics depends on the specific case and questions at hand. In many instances, combining different metrics (including beyond those discussed in this paper) likely will yield the most insights.

### 3.1 Robust timings and shifts

Directional statistics offers an advantage to evaluating streamflow seasonality across space and through time by eliminating the need to specify a start date for the water year, thereby producing stable and more robust timing results. In contrast, alternative metrics for determining streamflow timing, such as the half-flow date (as defined in Eq. 1) and the center of mass (as defined in Eq. 2), are dependent on an arbitrarily established start date for the water year. While some environments have clearly defined water years, establishing a consistent start date across diverse catchments and climates can be challenging whereby suitable starting dates vary regionally (Wasko et al., 2021; Sun & Cheruvelil, 2024). For example, most of Europe uses October 1$^{st}$ as the start of the water year, whereas November 1$^{st}$ is the norm in Germany (e.g., Renner et al., 2011). Note that in considering science questions that compare state or flux timing in the context of a water year, it is inherently required to define a start date, regardless of methodology. This may be particularly helpful in some instances, but in many other instances, a defined start date will be redundant or unnecessary information.

Whitfield (2013) already illustrated that the selection of a start date can problematically influence the inferred half-flow date (as defined in Eq. 1) and the center of mass (as defined in Eq. 2). We further demonstrate this with an example of a (simplified) flow regime before and after a temporal shift in streamflow (Fig. 2a). In this scenario, the original seasonal streamflow regime peaks after one-third of the year, while the shifted streamflow regime mirrors this but with its flows shifted one month earlier. Using directional statistics, the streamflow center of mass timing has a stable date (independent of the water-year start date). In this case, the date is 0.45 years since the start of the water year for the original streamflow regime and 0.37 years for the shifted streamflow regime. Thus, the center of mass date remains stable, and the temporal streamflow shift is fixed at a physically intuitive value of one month (Fig. 2c). In contrast, for the half-flow date (Eq. 1) and the center of mass (Eq. 2), the inferred central timing of streamflow shift at a different rate than changes in the water-year start date do. Consequently, the inferred timings become unstable (Fig. 2b). This instability indicates a high sensitivity of the inferred streamflow shift to the user's choice of start date (Fig. 2c), and the shift does not align with the intuitive one-month shift. We recognize that these exemplar streamflow regimes are simplified and that it would be illogical to select the beginning of the water year outside the

`

low-flow season; however, we highlight these examples to demonstrate that similar issues can and likely will arise when evaluating more complex streamflow regimes.

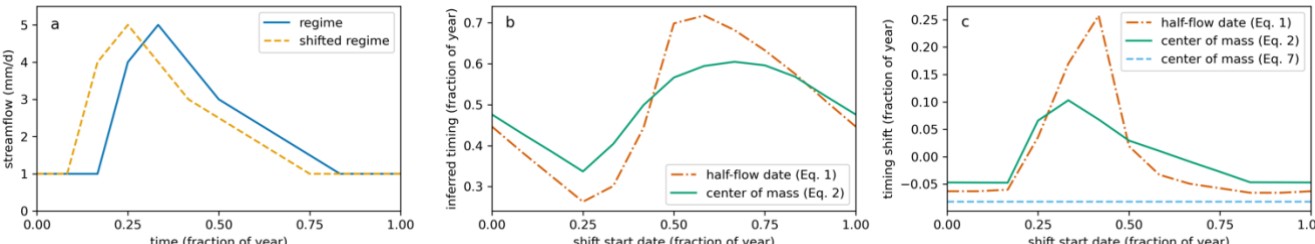

 **Figure 2. Effects of starting date of the water year on inferred streamflow timing. Panel (a) shows an example streamflow regime and a shifted regime, which is shifted in streamflow one month earlier. Panel (b) highlights instability in inferred central timing (using methodologies other than directional statistics, Eq. 1 and Eq. 2) of the original regime (not shown for the shifted regime) as the start date of the year is shifted forward. The shift in the selected start date (problematically) affects the inferred temporal streamflow shifts using Eqs. 1 and 2 but would remain stable as one-month change using directional statistics (c).**

 **3.2 Robust seasonality strengths**

Streamflow seasonality analyses using directional statistics assess the strength of seasonality based on the original temporal resolution of the dataset. For instance, when daily flow rates are provided, the timing and concentration can be derived from this daily data resolution. In contrast, most metrics that quantify the strength of seasonality (Eq. 3-5) require binning the data into an arbitrary time interval, typically per month (e.g., Feng et al., 2013; Wang et al., 2024; Oliver, 1980; Han et al., 2024).
 Binning is required as these methods do not account for the sequential nature of the (binned) timeseries, which prevents unambiguous differentiation between short-term variability and seasonal variations. However, binning comes with several limitations. A timeseries averaged into longer intervals loses a part of its temporal variability (e.g., Fig. 3) thereby changing the inferred seasonality strength when longer time intervals are used. Therefore, traditional seasonality metrics will show rates of seasonality that shift when longer binning intervals are used whereby it remains arbitrary which binning interval is
 appropriate, and this interval is potentially varying between catchments. In contrast, directional statistics do not require binning, and even when data are provided at a low resolution, streamflow seasonality's inferred strength remains mostly unaffected.

We exemplify this based on a 15-minute hydrograph for one year. For this hydrograph, we aggregate data to daily, weekly,
 and monthly resolution and calculate the strength of seasonality using traditional methods (Eqs. 3-5) and using directional statistics (Fig. 3). We also highlight the potential range of the seasonality indices based on a constant flow regime and a flow regime where all flow occurs at one single time interval (indicated in parentheses in Fig. 3); these ranges substantially change for all methods except directional statistics. In addition, while Eqs. 3-5 will infer a seasonality strength that strongly depends on the binning choice, directional statistics are insensitive to this process (and do not require binning). This sensitivity is
 present in the absolute value of seasonality strength, these seasonality strengths relative to their potential minimum and maximum, and the reference values determining the possible seasonality range. Besides these issues, note that not using

binning for Eqs. 3-5 is not an option, as it would shift the focus from quantifying seasonal fluctuations to short-term variability in streamflow.

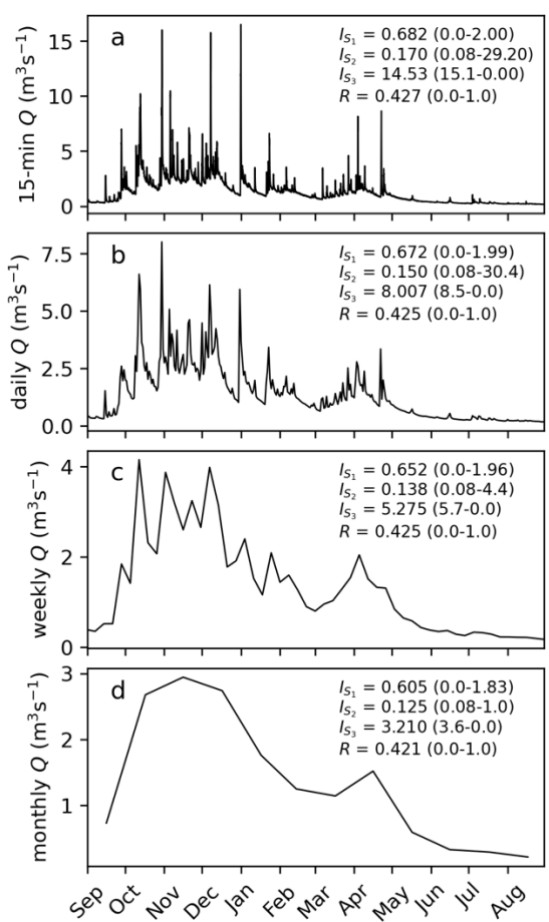

**Figure 3. Effects of binning interval on inferred seasonality strength. An example hydrograph, given at 15-min (a), daily (b), weekly (c), and monthly (d) resolution and the values of the associated seasonality metrics (Eqs. 3-5, and Eq. 8) with their possible range indicated in parentheses (constant regime – most seasonal regime). This indicates that traditional seasonality metrics tend to be more sensitive to the binning process than directional statistics are. This sensitivity is present in the absolute value of seasonality strength, the values relative to their potential minimum and**
**maximum, and the reference values determining the possible seasonality range. Note that binning is not required for directional statistics.**

Most methods used to quantify the strength of streamflow seasonality do not consider the sequential order of streamflow values. However, the sequences can determine the extent of seasonal bias within a flow regime. To illustrate this, we provide a simplified example of two regimes (Fig. 4) that, when analysed using the seasonality metrics defined in Eqs. 3-5 (which do

not consider the order of (monthly) flow values) would be assessed as having the same degree of seasonality. In contrast, when applying directional statistics to the concentration (Eq. 3), flow regime 1 is classified as seasonal ($R$=0.21), whereas flow

regime 2 is classified as non-seasonal ($R=0.0$) due to its identical flow rates across spring, summer, winter, and fall. In theory, strong seasonal variations in flow, with periodicities of six months or less, could occur and might be considered indicative of the seasonality of these flow regimes. However, such patterns are typically absent in measured streamflow time series (e.g., see Fig. 5 in Knoben et al., 2018). For other phenomena, such as bimodal precipitation regimes, directional statistics will struggle to characterize the bimodal nature of the annual cycle. It is important to note that the inferred strength of seasonality, as determined by Eqs. 3-5, is sensitive to the timing of the seasonal pattern. For example, in Fig. 4 (assuming constant rates within each month), any timing shift in the seasonal cycle would reduce the inferred seasonality by making monthly values more uniform. In contrast, directional statistics are not affected by this issue.

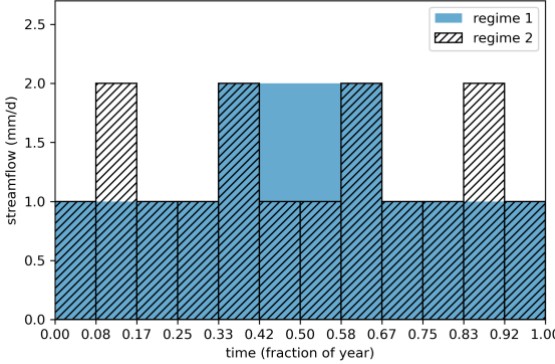

**Figure 4. Comparison of streamflow regimes illustrating differences in assessing the strength of seasonality. While both regimes appear to have the same degree of seasonality when analysed using traditional metrics (Eqs. 3-5), directional statistics quantify that flow regime 1 is seasonal ($R=0.21$), whereas flow regime 2 is non-seasonal ($R=0.0$) due to identical flow rates throughout the different seasons.**

## 4. Example applications

### 4.1 Water balances

The application of seasonality metrics can extend beyond streamflow. For example, directional statistics can characterize the seasonality of multiple water balance components within a catchment. Here, we illustrate this using the 43km$^2$ alpine Dischma catchment in Switzerland (mean elevation 2376 m.a.s.l.) (Fig. 5). In this catchment, precipitation is seasonal with higher rates in summer ($t_{\hat{P}} = 0.58$; $R = 0.28$) (Fig. 5a). A substantial fraction of annual precipitation falls as snow, as winter temperatures are below zero for part of the year, leading to highly seasonal snowpacks ($R=0.75$) with a center of mass at the end of winter ($t_{\hat{S}} = 0.20$) (Fig. 5b). Snowmelt from this snowpack is slightly less seasonally concentrated ($R =0.72$), with a center of mass in spring ($t_{\hat{M}} = 0.34$) (Fig. 5c). While most snow melts in spring, a smaller proportion of snowmelt also occurs during fall and early winter, slightly reducing the seasonal concentration of snowmelt. Energy availability in terms of potential evapotranspiration peaks in summer ($t_{\hat{E}_P}= 0.50$, $R = 0.39$) (Fig. 5d), and consequently evapotranspiration rates also have a distinct seasonality ($t_{\hat{E}} = 0.53$, $R =0.36$) (Fig. 5e). Streamflow remains low during the winter period (as no snowmelt occurs), rises following snow melts, and its center of mass occurs in early summer ($t_{\hat{Q}} = 0.51$) with substantial seasonality ($R = 0.46$)

(Fig. 5f). This approach enables us to track the evolution in the center of mass (and its strength) from snow to streamflow, highlighting seasonal interconnections between different water and energy fluxes.

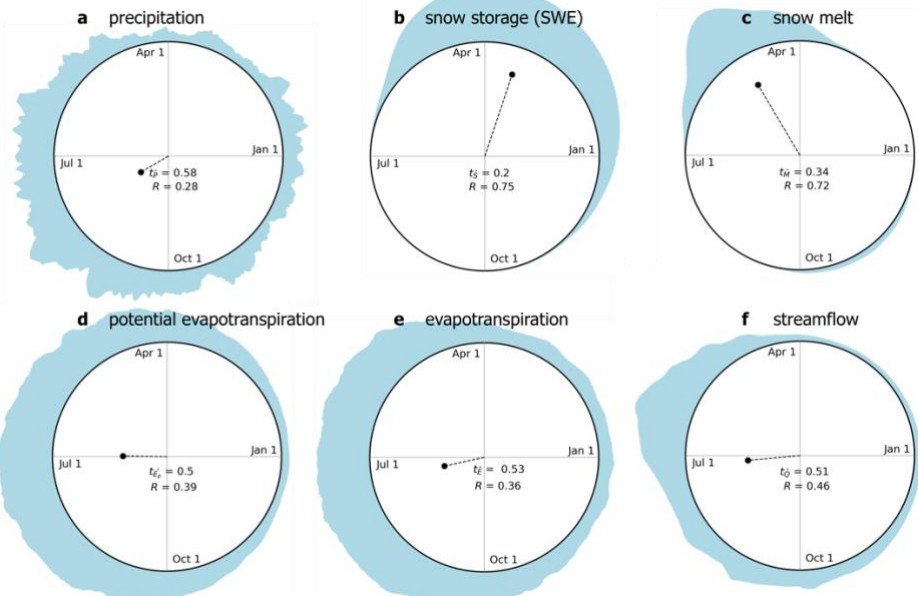

**Figure 5: Seasonality statistics of several water balance components of the Dischma catchment in Switzerland. Half-hourly precipitation and evapotranspiration are from the Davos flux site (Hörtnagl et al., 2023, data are available at https://www.swissfluxnet.ethz.ch/index.php/sites/site-info-ch-dav/). Streamflow data at the Kriegsmatte station is provided by the**
250 **Federal Office for the Environment (FOEN), and snow data from the Dischma station (MCH.DMA2) is provided by Meteoswiss. Streamflow and snow depth datasets are available in Magnusson et al. (2025). Snowmelt runoff are modelled at a daily timestep using operational snow-hydrological service (OSHD) model from 1998-2022 (Mott, 2023).**

## 4.2 Large-sample studies

A directional statistics approach to seasonality metrics can characterize the spatial gradients and regional differences in the
255 strength and timing of seasonal hydrological fluxes. To illustrate an example, we apply this approach to EStreams, a dataset of 17,130 European catchments (do Nascimento et al., 2024). First, we demonstrate regional seasonality differences by calculating the center of mass timing and concentration for catchments with at least 10 years of continuous streamflow data (11,117 stations).

Streamflow center of mass timing varies notably across Europe, and most variations are spatially highly autocorrelated. Winter-centered flows are widespread in the British Isles, coastal areas of the Baltic States, Denmark, much of Western Europe, Portugal, Extremadura and Andalucía (Spain), Italy, and Croatia (Fig. 6a), with varying degrees of seasonality strength (Fig. 6b). These winter-centered flows show the strongest seasonality in Portugal, southwestern Spain, Brittany (France), and a band in northern France stretching towards Luxembourg. Spring-centered flows are prevalent across much of Central and Eastern

Europe, southern Finland, and the pre-Alps (Fig. 6a), generally with weaker seasonality, such as in the northern Carpathians. Summer-centered flows are observed in the Alps, Norway, Sweden, and northern Finland (Fig. 6a), with the most pronounced seasonality in the Scandinavian Mountains and the higher regions of the Alps. These maps highlight broad-scale continental streamflow seasonality differences, likely largely driven by climate conditions, while also offering potential insights into the influence of landscape on these patterns. For instance, there is evidence of a geological signature in streamflow seasonality,

such as the locally delayed center of mass in Chalk aquifers in the Thames Basin (England) and northern France (Fig. 6a), or the stronger seasonality observed in Brittany and along a band of Jurassic rock in northern France (Fig. 6b). The center of mass timing (Eq. 7; Fig 6a) shows stronger regional timing differences than traditional metrics such as half flow date (Eq. 1) and center of mass (Eq. 2) (Figure A1).

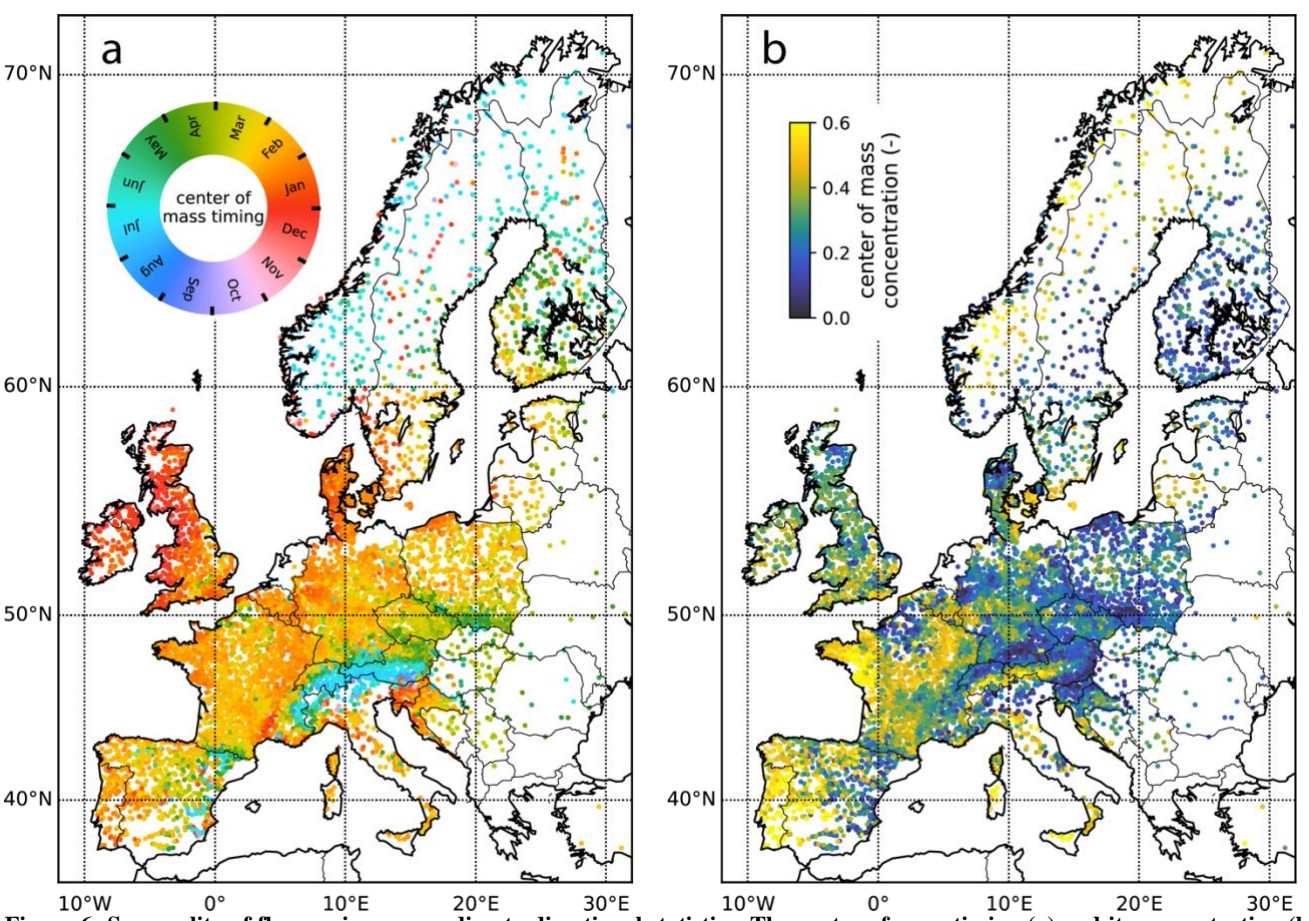

**Figure 6: Seasonality of flow regimes according to directional statistics. The center of mass timing (a) and its concentration (b) of catchments are shown at the location of the streamflow gauges and vary notably across Europe. Most of these variations are spatially highly autocorrelated. For illustration purposes, we do not show (the small number of) Icelandic catchments and streamflow gauges located East of 32ºE.**

`

### 4.3 Trend analyses

Directional statistics can be applied in trend analyses to evaluate whether the center of mass timing and the strength of the seasonal cycle have trends. The specifics of such trend analyses may vary depending on the exact research questions, the availability of data, and the selected trend estimator. Here, we demonstrate an example using the Theil-Sen estimator. Similar steps could be followed with other methods.

For each catchment, we calculate the center of mass timing values on an annual basis (Oct. 1 – Sep. 30) and assess possible trends using the Theil-Sen estimator:

$$\beta_{t_{\hat{Q}}} = \text{median}\left(\frac{t_{\hat{Q}_j} - t_{\hat{Q}_i}}{j - i}\right) \qquad \text{(Eq. 11)}$$

The trend estimator $\beta_{t_{\hat{Q}}}$ (year per year) is calculated as the median of the differences in dates across all possible year pairs ($i$ and $j$) based on annual values of $t_{\hat{Q}}$ (expressed as a fraction of the year). Such a trend analysis must account for the periodic nature of the year. We applied the unwrap function from NumPy (Harris et al., 2020) to $t_{\hat{Q}}$ to ensure that such discontinuities in $t_{\hat{Q}}$ are removed, creating a continuous representation of $t_{\hat{Q}}$ changes over time. Almost identical approaches have been applied to annual flood timings (e.g. Blöschl et al., 2017). These approaches do not require an unwrap function but assume two dates (e.g., $t_{\hat{Q}_j} - t_{\hat{Q}_i}$) cannot be more than 0.5 year apart. However, this assumption can sometimes violate the time's arrow in trend analyses. The Theil-Sen estimator of the concentration $\beta_R$ (1/year) can be calculated as:

$$\beta_R = \text{median}\left(\frac{R_j - R_i}{j - i}\right) \qquad \text{(Eq. 12)}$$

The trend estimator $\beta_R$ (1/year) is calculated as the median of the differences in dates across all possible year pairs ($i$ and $j$) based on annual values of $R$. To conduct these trend analyses, we use the `theilslopes` function from the `scipy.stats` module in SciPy (Virtanen et al., 2020).

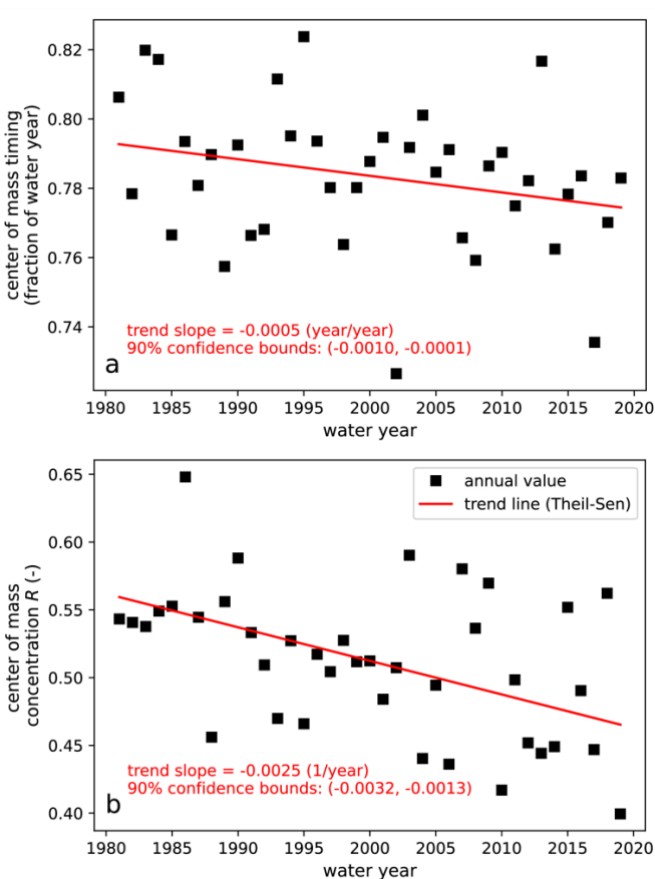


**Fig. 7. Trend analysis of the center of mass timing (a) and its concentration (b) for streamflow in the Dischma catchment. The 90% confidence interval reflects the uncertainty in the estimated slope. Over the period 1980-2020, center of mass of streamflow has shifted earlier in the year with weaker seasonality.**

For instance, trend analysis of the streamflow in the Dischma catchment (same gauge as in Fig. 5) reveals that the center of mass has shifted earlier in the year at a rate of 0.0005 years (or 0.18 days) per year from 1980 to 2020 (90% confidence bounds: -0.0010, -0.0001) (Fig. 7a), while the concentration has decreased at a rate of 0.0025 years per year (90% confidence bounds: -0.0032, -0.0012) (Fig. 7b). This suggests that the streamflow has shifted earlier in the year (7.3 days total) with less seasonality. Note that such analysis could be applied to a larger dataset or to ask detailed process questions. Such application

is not provided as this should be separate studies and not part of this technical note.

## 5 Summary

Streamflow and other hydrological fluxes typically vary across seasons. Existing metrics designed to characterize seasonality

often do not account for the periodic nature of seasonal cycles. Here, we use directional statistics to apply the concept of the center of mass for unbounded environments. This approach allows for the simultaneous quantification of seasonal timing

`

(*center of mass timing*) and strength (*center of mass concentration*). We demonstrate that using directional statistics provides a more mathematically robust quantification of seasonality compared to several widely used seasonality metrics. To illustrate its application, we analyse data from European catchments, showcasing the method's utility for various water balance 320 components, large-sample hydrological studies, and trend analyses. The introduced metrics, leveraging directional statistics, offer tools for studying the seasonality of environmental fluxes both within and beyond hydrology.

## Appendix A1: Methodological Equations, Discrete form

In discrete form, the *center of mass timing* is calculated as, $t_{\hat{Q}}$ [T]:

$$t_{\hat{Q}} = \frac{\text{atan2}(\overline{y}, \overline{x})}{2\pi} \tag{Eq. A1}$$

and its *concentration, R* [dimensionless]:

$$R = \sqrt{\overline{x}^2 + \overline{y}^2} \tag{Eq. A2}$$

where the cosine and sine components of streamflow are:

$$\overline{x} = \frac{1}{\sum_{i=1}^{n} Q_i} \sum_{i=1}^{n} \cos(2\pi t_i) \times Q_i \tag{Eq. A3}$$

$$\overline{y} = \frac{1}{\sum_{i=1}^{n} Q_i} \sum_{i=1}^{n} \sin(2\pi t_i) \times Q_i \tag{Eq. A4}$$

where $Q_i$ is the streamflow rate at interval *i* [L/T or L$^3$/T], *n* is the total number of intervals considered, and $t_i$ expresses the timing at interval *i* [years since Jan. 1].

`

**Appendix A2: Traditional seasonality metrics**

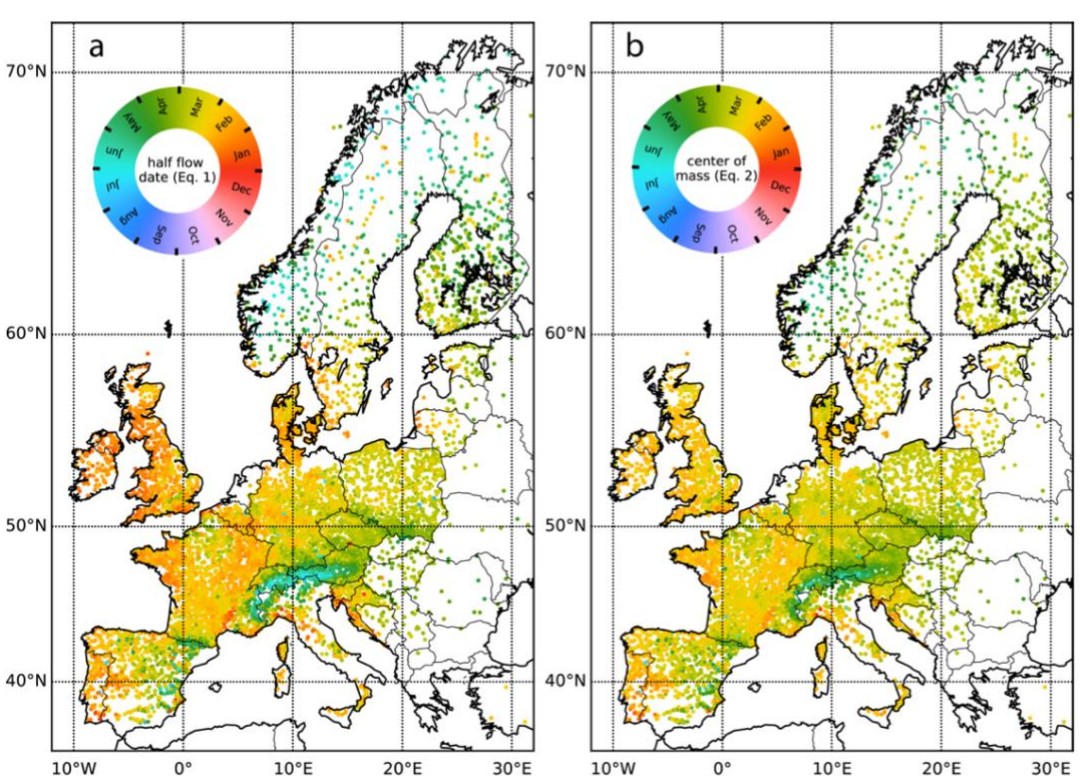

**Figure A2: Seasonal timing of flow regimes (without directional statistics) according to the half flow date (Eq. 1) and center of mass (Eq. 2) based on a water year starting Oct 1st.**

**Data availability.** Data is available via the cited sources.

**Author contributions.** W.R.B.: methods, analysis, writing (original draft preparation). K.H.: methods, writing (reviewing and editing). H.B.: methods, writing (reviewing and editing).

**Competing interests.** The contact author has declared that none of the authors has any competing interests.

**Acknowledgements.** The authors thank Sebastian Carugati, Anna Luisa Hemshorn de Sánchez, Mira Anand, and the reviewers for their valuable suggestions.

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
