# Peer review of "Technical Note: Streamflow Seasonality using Directional Statistics"

_EGUsphere, 2024_

## Author Comment (AC4)

**We thank the reviewer for further clarifying this and provide information below that will be added to the paper.**

Thanks to the authors for the prompt response. It is fine to add comparisons in the discussion part. However, I think adding figures based on non-directional statistics will still be useful. Showing the problems is as valuable as showing the advantages. You can adpot the normal Oct-Sep water year, even though Germany may use a different norm, and combine the trend and spatial distributions in one figure to avoid extended length. Such kind of comparison will be necessary to validate the distinct features of the directional statistics and show how problematic the non-directional statistics could be if those problems do exist. About the unit problem, its impact won't be not huge if you aim to compare the spatial patterns and trends. There is no need to create differences or calculate metrics between those statistics.

**We will describe this in text and add a figure to the paper that includes these dates according to the half flow date (Eq. 1) and center of mass (Eq. 2) based on a water year starting Oct 1st. This approach makes dates more homogenized across Europe (see Fig. below). We will point this out and discuss their relative patterns and interpretations.**

[Figure]

**Fig. Seasonal timing of flow regimes (without directional statistics) according to the half flow date (Eq. 1) and center of mass (Eq. 2) based on a water year starting Oct 1st.**

---

## Author Response (AR1)

Dear Editor,

We thank the Editor, the two reviewers, and Gabriele Villarini (**CC1**) for these constructive and positive evaluations. Below we provide a point-by-point response to reviewer comments, and highlight the changes made to the manuscript.

We also suggest making a slight adjustment to the terminology. Previously, we referred to timing of streamflow (using *center of mass*) and the strength of its seasonal cycle (using *concentration*). Upon further thought, we believe that (a clearer representation that better highlights these variables are connected) by referring to them as the *center of mass timing* and the *center of mass concentration*. This better emphasizes the center of mass is the mean x- and y-coordinates combination, and that it can be summarized through two dimensions (timing and concentration).

On behalf of all authors,

Wouter Berghuijs

==Editor evaluation== **(Our responses in bold)**

Dear Authors,
Thank you for your submission. Your manuscript has been reviewed by experts in the field, and while they recognize its contributions, they have identified several key issues that require major revision before further consideration for publication.

**We fully address all these issues below**

The reviewers appreciate the study's approach and its efforts to generalize previous methods. However, they have highlighted the need for additional comparisons with non-directional metrics, clearer mathematical representation of equations, stronger integration of relevant literature, clarification on timing comparisons, and normalization of seasonality strength metrics.

**The revised manuscript now includes extra comparisons with non-directional metrics (with a new figure), clearer mathematical representation of equations (the equations are now also provided in discrete form), stronger integration of relevant literature (we now cite three studies to acknowledge these contributions, including one reference not provided by the reviewers), clarification on timing comparisons, and normalization of seasonality strength metrics (we also make normalized comparison, and provide extra textual clarification). All these comments are detailed below in the responses to the reviewers.**

Additionally, they recommend improving the clarity and organization of the manuscript to enhance readability and impact.

**We addressed all reviewer comments below, which improve both the clarity and organization of the manuscript.**

I recommend that you carefully address these concerns and submit a revised manuscript. Your revision will undergo further review to determine its suitability for publication.
I appreciate your contributions to the field and look forward to receiving your revised manuscript.
Best regards,
Fuqiang

**Thank you, we look forward to the new round of reviews.**

CC1: 'Comment on egusphere-2024-4117', Gabriele Villarini, 27 Jan 2025

**(Our responses in bold, here identical to what we already posted on EGUSPHERE-2024-4117) and new changes to the manuscript in red**

I have given a quick look at the manuscript, and believe that authors may have missed several relevant papers that already get to some of the core of their arguments and actually move them forward:

- Barth, Nancy A., et al. "Changes in streamflow seasonality associated with hydroclimatic variability in the north-central United States among three discrete temporal periods, 1946–2020." Journal of Hydrology: Regional Studies 57 (2025): 102084.
- Dhakal, Nirajan, et al. "Nonstationarity in seasonality of extreme precipitation: A nonparametric circular statistical approach and its application." Water Resources Research 51.6 (2015): 4499-4515.
- Treppiedi, D., G. Villarini, J. Bender, and L.V. Noto, Precipitation extremes projected to increase and to occur in different times of the year, Environmental Research Letters, 20(1), 014014, 2025.
- Veatch, W., and G. Villarini, Modeling riverine flood seasonality with mixtures of circular probability density functions, Journal of Hydrology, 613, 1-11, 2022.
- Villarini, G., On the seasonality of flooding across the continental United States, Advances in Water Resources, 87, 80-91, 2016.

This is not an exhaustive list and I hope these suggestions will help the authors to better contextualize their results with respect to the broader literature.

**We appreciate the reminder that many studies use circular/directional statistics to characterize the seasonality of extremes, as evidenced by the suggested references. In our manuscript, we acknowledge this development by stating:**

***"Directional statistics have been widely used to characterize the seasonality of extreme flows and extreme precipitation (e.g., Burn et al., 1997; Young et al., 2000; Merz & Blöschl, 2003; Laaha and Blöschl, 2006; Dhakal et al., 2015; Villarini, 2016; Blöschl et al., 2017; Berghuijs et al., 2016, 2019; Floriancic et al., 2021; Chagas et al., 2022)."***

**We have incorporated two of the suggested references, although an exhaustive list of all studies that could be cited here would be far more extensive than what is provided by us and these suggestions.**

**Importantly, our manuscript develops and applies directional statistics to characterize (continous) seasonal flow regimes, rather than focusing on the seasonality of annual extremes or peaks-over-threshold *events*. Our approach thereby unifies the concepts of seasonal flow regimes, center of mass, and directional statistics—which is not covered by the referenced studies. These developments are probably not complicated to someone familiar with directional statistics for characterizing extreme events, but have not been shown in the (suggested) literature, and do have many potential applications (also beyond hydrology).**

Thank you for the quick reply. I appreciate the clarification and found this technical note interesting and well-suited for this venue.

**We thus consider this issue to be resolved**

RC1: 'Comment on egusphere-2024-4117', Anonymous Referee #1, 09 Feb 2025

**(Our responses in bold, here identical to what we already posted on EGUSPHERE-2024-4117), and new reponses in green, and new changes to the manuscript in red**

This study proposed a new metric based on directional statistics to quantify the streamflow seasonality, which is a significant contribution of hydrology since the streamflow seasonality is an important characteristic of hydrograph. The paper is overall well written and organized. However, I would like to post some questions and suggestions regarding the applicability, novelty and robustness of the proposed metrics. The major concerns are as followed:

**We thank the reviewer for this constructive report and address these individual points below**

1. Could the authors provide a discrete form of the calculation equation in addition to the integral form? This would make the adopting of the equation more practical, since the streamflow data is always in discrete form, such as daily.

**We added a discrete form of these equations in the revised manuscript.**

**Appendix A1: Methodological Equations, Discrete form**

In discrete form, the *center of mass timing* is calculated as, $t_{\hat{Q}}$ [T]:

$$t_{\hat{Q}} = \frac{\text{atan2}(\overline{y}, \overline{x})}{2\pi} \qquad \text{(Eq. } A1)$$

and its *concentration*, $R$ [dimensionless]:

$$R = \sqrt{\overline{x}^2 + \overline{y}^2} \qquad \text{(Eq. } A2)$$

where the cosine and sine components of streamflow are:

$$\overline{x} = \frac{1}{\sum_{i=1}^{n} Q_i} \sum_{i=1}^{n} \cos(2\pi t_i) \times Q_i \qquad \text{(Eq. } A3)$$

$$\overline{y} = \frac{1}{\sum_{i=1}^{n} Q_i} \sum_{i=1}^{n} \sin(2\pi t_i) \times Q_i \qquad \text{(Eq. } A4)$$

Here, $Q_i$ is the mean streamflow rate at interval $i$ [L/T or L$^3$/T], $n$ is the total number of intervals considered, and $t_i$ expresses the timing at interval $i$ [years since Jan. 1].

1. I've seen a similar metric called concentration ratio (CR) and concentration period (CP) in some studies (e.g., Nan & Tian 2024; Jiang et al., 2022). If I understand correctly, CR and CP seems to be a simplified form of the R and tQ proposed in this study, which only uses the monthly data. I think the authors can conducted more literature reviews to illustrate the origin and the novelty of the proposed metric.

**We indeed did not notice the interesting works by Nan & Tian (2024) and Jiang et al. (2022).**

**In these works, a simplified form of mass center and concentration are provided, and we now explicitly acknowledge these in our revised manuscript.**

**We, however, believe that providing a more generalized and transferrable form, including several applications, and robustness tests, can catalyze its uptake in hydrological (and other fields') research.**

Recent developments show these can be adapted to represent the overall intra-annual distribution of streamflow (Jiang et al., 2022; Nan and Tian, 2024; Hanus et al., 2024).

2. The authors illustrate that tQ remains stable regardless of the shift start data, which is indeed an advantage. However, it seems that when interpreting the calculated tQ, we still cannot clealy understand the seasonality pattern if we don't know the start date of water year. For example, if the tQ of catchment A and B are 0.4 and 0.5, we cannot say the mass center of A catchment is earlier than B catchment in the corresponding water year if we don't know the start date.

**For questions that consider a temporal analysis within the context of a water year, it is inherently required to defineine a start date, regardless of methodology. In some instances, this may be particularly helpful.**

**Yet in other instances, for example, to understand mass center shifts or differences between places (either in space or time), one can compare differences in the timing, irrespective of the water year start.**

**We have better emphasized both aspects in the revised manuscript.**

**L158-160:** Note that in considering science questions which compare state or flux timing in the context of a water year, it is inherently required to define a start date, regardless of methodology. This may be particularly helpful in some instances, but in many other instances, a defined start date will be redundant or unnecessary information.

3. Regarding the robustness of seasonality strengths, a normalization based on the maximum variation range of each metric is needed. Otherwise, for the example in Figure 3, it's difficult to say whether the change of Is3 from 14.53 to 3.21 is more significant than the change of R from 0.427 to 0.421.

**We now provide extra analyses and an updated figure. These further highlight the advantages of directional statistics over the other methods.**

**L199-211:** We exemplify this based on a 15-minute hydrograph for one year. For this hydrograph, we aggregate data to daily, weekly, and monthly resolution and calculate the strength of seasonality using traditional methods (Eqs. 3-5) and using directional statistics (Fig. 3). We also highlight the potential range of the seasonality indices based on a constant flow regime and a flow regime where all flow occurs at one single time interval (indicated in parentheses in Fig. 3); these ranges substantially change for all methods except directional statistics. In addition, while Eqs. 3-5 will infer a seasonality strength that strongly depends on the binning choice, directional statistics are insensitive to this process (and do not require binning). This sensitivity is present in the absolute value of seasonality strength, these seasonality strengths relative to their potential minimum and maximum, and the reference values determining the possible seasonality range. Besides these issues, note that not using binning for Eqs. 3-5 is not an option, as it would shift the focus from quantifying seasonal fluctuations to short-term variability in streamflow.

[Figure]

Figure 3. Effects of binning interval on inferred seasonality strength. An example hydrograph, given at 15-min (a), daily (b), weekly (c), and monthly (d) resolution and the values of the associated seasonality metrics (Eqs. 3-5, and Eq. 8) with their possible range indicated in parentheses (constant regime –most seasonal regime). This indicates that traditional seasonality metrics tend to be more sensitive to the binning process than directional statistics are. This sensitivity is present in the absolute value of seasonality strength, the values relative to their potential minimum and maximum, and the reference values determining the possible seasonality range. Note that binning is not required for directional statistics.

Nan, Y., & Tian, F. (2024). Glaciers determine the sensitivity of hydrological processes to perturbed climate in a large mountainous basin on the Tibetan Plateau. Hydrology and Earth System Sciences, 28(3), 669-689.

Jiang, Y., Xu, Z., & Xiong, L. (2022). Runoff variation and response to precipitation on multi-spatial and temporal scales in the southern Tibetan Plateau. Journal of Hydrology: Regional Studies, 42, 101157.

**No further comments.**

**We thank the Reviewer for this helpful review.**

RC3: 'Comment on egusphere-2024-4117', Anonymous Referee #2, 02 Mar 2025

**(Our responses in bold, here identical to what we already posted on EGUSPHERE-2024-4117), and changes to the manuscript in red (which are new)**

**We thank the reviewer for this positive and constructive evaluation**

This study proposes the use of directional statistics to characterize both the timing and strength of the seasonal streamflow cycle. It begins with a series of idealized comparisons to illustrate the methodology and then applies the statistics to real-world streamflow data. While prior studies (e.g., Nan & Tian, 2024; Jiang et al., 2022, indicated by another reviewer) have employed similar forms of directional statistics, this work advances the field by presenting a more generalized framework that is applicable to any temporal resolution. Additionally, it provides a comprehensive discussion on the advantages of directional statistics, including its ability to robustly capture seasonal timings and shifts (Section 3.1) and seasonal strength (Section 3.2). I believe this study makes a valuable contribution to the field of streamflow seasonality research.

**Thank you.**

I have only one suggestion for further improvement. Sections 4.1–4.3 effectively demonstrate the application of directional statistics. However, it would be highly informative to include a comparison with non-directional metrics, similar to what was done in Figures 1 and 2. Such a comparison would provide deeper insights into the practical differences between these metrics in streamflow analysis. In particular, I would be interested in seeing these comparisons extended to Figures 6 and 7, as they would help readers better understand the unique advantages of directional statistics in real-world scenarios.

**We could provide a comparison with non-directional statistics for timing using the half-flow date and the center of mass (Eqs. 1 and 2). However, displaying these metrics in maps in the main paper seems problematic, as these new maps would lack physical meaning. This is because either the metrics use different starting dates of the water year regionally (and then the units of the metric differ, which makes them incomparable) or they begin at unrealistic times of the year (e.g., not matching the low flow season) which is not informative either. Consequently, we are hesitant to present these metrics in a Figure, as they may distract from the more meaningful examples shown (which is the purpose of Sections 4.1-4.3). Instead, we propose addressing this issue through a textual discussion in Sections 4.2 and 4.3.**

Thanks to the authors for the prompt response. It is fine to add comparisons in the discussion part. However, I think adding figures based on non-directional statistics will still be useful. Showing the problems is as valuable as showing the advantages. You can adpot the normal Oct-Sep water year, even though Germany may use a different norm, and combine the trend and spatial distributions in one figure to avoid extended length. Such kind of comparison will be necessary to validate the distinct features of the directional statistics and show how problematic the non-directional statistics could be if those problems do exist. About the unit problem, its impact won't be not huge if you aim to compare the spatial patterns and trends. There is no need to create differences or calculate metrics between those statistics.

**We thank the reviewer for further clarifying this and provide information below that is now added to the paper.**

Thanks to the authors for the prompt response. It is fine to add comparisons in the discussion part. However, I think adding figures based on non-directional statistics will still be useful. Showing the problems is as valuable as showing the advantages. You can adpot the normal Oct-Sep water year, even though Germany may use a different norm, and combine the trend and spatial distributions in one figure to avoid extended length. Such kind of comparison will be necessary to validate the distinct features of the directional statistics and show how problematic the non-directional statistics could be if those problems do exist. About the unit problem, its impact won't be not huge if you aim to compare the spatial patterns and trends. There is no need to create differences or calculate metrics between those statistics.

**We describe this in text and have added a figure to the paper that includes these dates according to the half flow date (Eq. 1) and center of mass (Eq. 2) based on a water year starting Oct 1$^{st}$. This approach**

**makes dates more homogenized across Europe (see Fig. below). We will point this out and discuss their relative patterns and interpretations.**

**L370-272:** The center of mass timing (Eq. 7; Fig 6a) shows stronger regional timing differences than traditional metrics such as half flow date (Eq. 1) and center of mass (Eq. 2) (Figure A1).

**Appendix A2:**

[Figure]

Figure A1: Seasonal timing of flow regimes (without directional statistics) according to the half flow date (Eq. 1) and center of mass (Eq. 2) based on a water year starting Oct 1st.

---

## Author Response (AR2)

Dear Editor,

We thank you and the reviewers for this evaluation.

We addressed the final concern of Reviewer #1 (see below)

We have now submitted all files required for production.

On behalf of all authors,

Wouter Berghuijs
* * *
**Comment Reviewer #1**

I appreciate the authors' efforts in revising the manuscript. All my comments have been addressed in this version. However, I still have some minor concerns regarding the robustness of the metrics. Based on the updated Figure 3, which illustrates the ranges of each metric, it appears that, except for IS2, the other three metrics are not highly sensitive to the binning choice. For instance, IS1 is 34.1% of its maximum when using 15-minute data and decreases slightly to 33.1% when using monthly data, which corresponds to 97.1% of the value calculated with 15-minute data. Similarly, for the directional metric, the corresponding ratio is 98.6%, only slightly higher than that of IS1, indicating that its sensitivity is only marginally weaker than IS1. I am not questioning the usefulness of the directional metric; rather, I believe that the differences among the various metrics should be conveyed more accurately. At the very least, I do not think IS1 and IS3 "strongly depend on the binning choice." Furthermore, instead of providing absolute values and potential ranges, I suggest presenting the relative values directly, as this would facilitate a clearer understanding of the differences among the metrics.

**We changed the text and figure as follows:**

**3.2 Robust seasonality strength**

Streamflow seasonality analyses using directional statistics assess the strength of seasonality based on the original temporal resolution of the dataset. For instance, when daily flow rates are provided, the timing and concentration can be derived from this daily data resolution. In contrast, most metrics that quantify the strength of seasonality (Eq. 3-5) require binning the data into a longer time interval, typically per month (e.g., Feng et al., 2013; Wang et al., 2024; Oliver, 1980; Han et al., 2024). However, these approaches that rely on binning come with several limitations. First, the theoretical range of seasonality values depends on the binning timescale. Second, Eqs. 3-5 do not account for the sequential nature of the (binned) timeseries, which prevents unambiguous differentiation between short-term variability and seasonal variations. Third, selecting a binning interval length is inherently arbitrary, and, as just stated, this decision impacts both the attainable range of seasonality values and the degree to which the method captures short-term versus seasonal variations.

We exemplify such limitations based on a 15-minute hydrograph for one year. For this hydrograph, we aggregate data to daily, weekly, and monthly resolution and calculate the strength of seasonality using traditional methods (Eqs. 3-5) and using directional statistics (Eq. 8) (Fig. 3). We also highlight the potential range of the seasonality indices based on a constant flow regime and a flow regime where all flow occurs at one single time interval (indicated in parentheses in Fig. 3) and the seasonality compared to its theoretical maximum value (expressed in percentage). Note that for the entropy-based measure, stronger seasonality is associated with lower entropy values. A timeseries binned into longer intervals loses a part of its temporal variability and shifts its potential seasonality range (Fig. 3). As a result, the absolute and the relative inferred seasonality strengths shift depending on the binning. For some methods, these changes are very large (e.g., Eqs. 4), whereas for other methods changes are smaller (e.g. Eqs. 3 and 5). However, also in these latter cases, inferred seasonality strengths do not unambiguously differentiate between short-term variability and seasonal variations. Directional statistics are insensitive to these problems (and do not require binning) and thereby have relatively constant absolute seasonality and potential seasonality ranges, and quantify seasonal variations without the degree of short-term variations substantially affecting the inferred seasonality strength.

[Figure]

Figure 3. Effects of binning interval on inferred seasonality strength. An example hydrograph, given at 15-min (a), daily (b), weekly (c), and monthly (d) resolution and the values of the associated seasonality metrics (Eqs. 3-5, and Eq. 8) with their possible range indicated in parentheses (constant regime – most seasonal regime) and the seasonality compared to its theoretical maximum value (expressed as a percentage). This indicates that traditional seasonality metrics tend to be more sensitive to the binning process than directional statistics are. This sensitivity is present in the absolute value of seasonality strength, the values relative to their potential minimum and maximum, and the reference values determining the possible seasonality range. Note that binning is not required for directional statistics.